# Two-Stage Vision Transformers and Hard Masking offer Robust Object Representations

## Abstract

Context can strongly affect object representations, sometimes leading to undesired biases, particularly when objects appear in out-of-distribution backgrounds at inference. At the same time, many object-centric tasks require to leverage the context for identifying the relevant image regions. We posit that this conundrum, in which context is simultaneously needed and a potential nuisance, can be addressed by an attention-based approach that uses learned binary attention masks to ensure that only attended image regions influence the prediction. To test this hypothesis, we evaluate a two-stage framework: stage 1 processes the full image to discover object parts and identify task-relevant regions, for which context cues are likely to be needed, while stage 2 leverages input attention masking to restrict its receptive field to these regions, enabling a focused analysis while filtering out potentially spurious information. Both stages are trained jointly, allowing stage 2 to refine stage 1. The explicit nature of the semantic masks also makes the model's reasoning auditable, enabling powerful test-time interventions to further enhance robustness. Extensive experiments across diverse benchmarks demonstrate that this approach significantly improves robustness against spurious correlations and out-of-distribution backgrounds. Code is available in this anonymized repository

## 1 Introduction

Deep learning models often rely on contextual cues to learn object representations. While this can be beneficial for certain tasks, it can also introduce spurious correlations on which the model learns to rely, hampering generalization Rosenfeld et al. (2018); Choi et al. (2012); Xiao et al. (2021). A common example is when models prioritize background cues over intrinsic object properties, leading to failures in out-of-distribution (OOD) settings where such correlations no longer hold Aniraj et al. (2023), something that happens often in practice Beery et al. (2018). It is therefore crucial to ensure that the model focuses on task-relevant image regions.

Models that integrate spatial attention maps directly into their inference process can help guiding the model towards focusing on relevant image regions and have the potential to provide guarantees of faithfulness, as they reveal the reasoning of the model. Typically, such methods apply a learned soft attention mask to a high-level feature map : a technique we refer to as **late masking**. However, this approach is undermined by two distinct sources of information leakage that limit robustness. First, the *late* application of the mask means that deep features are already contaminated by background information due to the vast receptive fields of modern architectures. Second, the *soft*, non-binary nature of the attention masks assigns non-zero weights to all locations, allowing for further, residual leakage from spurious regions. This combination of flaws, illustrated in Fig. 1 (top), critically undermines the model's ability to ignore spurious cues.

To truly prevent reliance on spurious cues, we posit that a solution is a model architecturally blind to them. We study a two-stage framework with **early masking** that provides this architectural guarantee. Building on a recent part-discovery method Aniraj et al. (2024), our Stage 1 (Selector) processes the full image to generate a discrete, binary mask identifying relevant foreground regions. Subsequently, our Stage 2 (Predictor), a second Vision Transformer, receives only the input tokens corresponding to this foreground mask. By operating on this subset of the input, its receptive field is strictly constrained, making it physically impossible to access or exploit information from the masked-out background regions. The two stages are trained jointly, allowing the downstream task

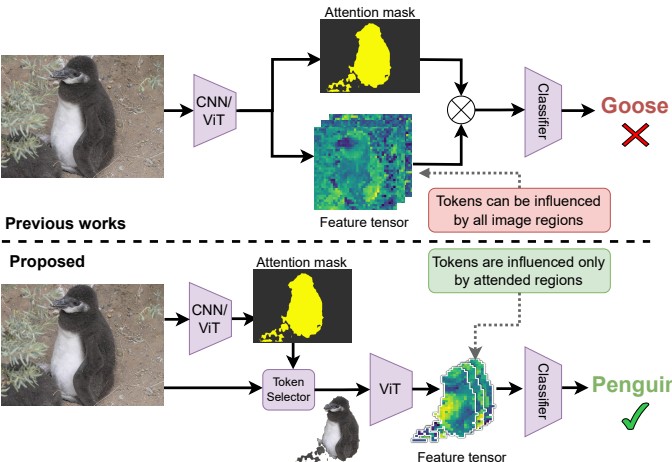

Figure 1: Previous attention-based approaches apply the attention mask to a deep feature tensor, where all locations can be affected by the whole image due to large receptive fields (top). Our approach ensures that only the selected tokens contribute to the downstream task (bottom).

to refine the foreground selection. This design ensures the model learns using *only* the object itself, leading to state-of-the-art results on several challenging robustness benchmarks. Furthermore, the explicit semantic masks make the model's reasoning auditable and enable powerful test-time interventions to further enhance performance.

## 2 RELATED WORKS

**Spatial attention in computer vision.** Attention mechanisms induce the model to focus on a subset of the input that is deemed relevant to the task at hand. Originally introduced to reduce computational load in image classification Mnih et al. (2014), spatial attention gained traction in captioning Xu et al. (2015), visual reasoning Hudson & Manning (2018), and other tasks Guo et al. (2022) where focusing on key image regions allows the model to decompose the complex task into multiple, simpler ones. Recent work on part discovery Huang & Li (2020); van der Klis et al. (2023); Aniraj et al. (2024) also leverages attention mechanisms. These approaches assume that focusing the attention on the correct parts will lead to better classification results, and leverage this learning signal to discover the semantic parts that compose the objects of interest. However, this paradigm is fundamentally limited by two sources of information leakage: they perform **late masking** on deep features already contaminated by background cues due to large receptive fields, and they use **soft attention masks** that allow for residual leakage from all image regions. This can potentially reduce *faithfulness*, or the degree to which only task-relevant image areas are effectively accessible. This concern has led to work measuring the faithfulness of attention maps in ViTs Wu et al. (2024b), as well as improving it Xie et al. (2022); Wu et al. (2024a); Ntrougkas et al. (2024). Our work proposes an architectural solution that directly addresses these sources of leakage.

**Local object representations.** Object-centric computer vision tasks require representations that remain invariant to changes in backgrounds and co-occurring objects. Previous works provide local object representations via mask-invariance losses Stone et al. (2017), clustering-like losses Yun et al. (2022) or directly altering the attention mechanism Ibtehaz et al. (2024). While some methods aim to align post-hoc explanations with segmentation maps Ross et al. (2017), they do not guarantee that only attended areas contribute to the decision, with studies highlighting information contamination from outside the object attention masks due to large receptive fields Aniraj et al. (2023).

**Input attention maps for interpretability.** Auxiliary mask predictors have been proposed to explain black-box classifiers by identifying minimal masks that preserve predictions without retraining Yuan et al. (2020); Phang et al. (2020); Stalder et al. (2022); Brinner & Zarrieß (2023); Zhang et al. (2024). Others use *post hoc* attribution maps to guide training Ismail et al. (2021). Closer to

our approach, joint amortized explanation methods (JAMs) Chen et al. (2018); Yoon et al. (2018); Ganjdanesh et al. (2022) jointly learn selector and predictor models. However, a key limitation of these methods is that a selector driven only by a simple classification loss can fail to distinguish causal features from spurious ones, or even encode class information directly into the selection pattern Jethani et al. (2021); Puli et al. (2024). Some of the proposed solutions to this problem involve either unstructured selection masks Jethani et al. (2021) or simplistic ones parametrized as a single spatial Gaussian Ganjdanesh et al. (2022). More recently, COMET Zhang et al. (2024) proposed to aim at finding the complete foreground, rather than just a sufficient mask. We explore the suitability of solving this by integrating part-shaping losses into the selector while seeking compatibility with vision transformers, and focus our evaluation on the impact on robustness against OOD backgrounds. Furthermore, this semantic part-based approach enables a unique capability absent in prior work: auditable, test-time interventions.

**Input attention maps for robustness.** Joint learning of input masks has also been explored to enhance robustness. Xiang et al. (2021) shows that limiting the receptive field and applying targeted patch masking improves adversarial robustness. Spurious correlations can be mitigated by isolating foreground regions and building image composites with mismatched backgrounds Xiao et al. (2023); Noohdani et al. (2024); Chakraborty et al. (2024), encouraging the model to rely on foreground cues. Asgari et al. (2022) masks key image regions using attribution maps, forcing the model to identify alternative features and assess potential spurious correlations. Multiple spurious cues can coexist in a dataset, and techniques designed to mitigate one may inadvertently amplify another Li et al. (2023). In this work, we leverage part discovery to simultaneously model several of these correlations.

**Token Pruning and Sparse Attention.** Efficiency-oriented approaches such as sparse attention Wei et al. (2023); Zhu et al. (2023) or token pruning Tang et al. (2023); Rao et al. (2021); Chen et al. (2021) select tokens mainly to speed up inference based on task discriminativeness. However, similar to JAMs, a selector guided solely by task discriminativeness can inadvertently learn to prioritize spurious-but-predictive features over causal ones. In contrast, iFAM's token selection is driven by a joint objective that couples classification with part-shaping losses to discover semantically consistent foreground regions that are also useful for solving the downstream task.

## 3 METHODOLOGY

**iFAM** (**I**nherently **F**aithful **A**ttention **M**aps for vision transformers) depicted in Fig. 2, consists of two stages: the first one has access to the whole image and predicts which image regions should be selected for the second stage. These selected regions then define the receptive field used by the second stage for solving the downstream task. This design ensures that the second stage can only pay attention to the selected image regions, guaranteeing that it cannot make use of any information outside the mask.

### 3.1 EARLY VS LATE MASKING

**Existing attention-based methods (Late Masking)** typically learn two functions on the input: a selector that produces a mask, and a feature extractor. An image feature vector is then computed by masking the high-level features from the feature extractor. If we denote the selector model as $h_{\theta_s}(\cdot)$ and the feature extractor as $h_{\theta_p}(\cdot)$, this process can be described as:

$$\mathbf{y} = g_\phi(h_{\theta_p}(\mathbf{x}) \odot h_{\theta_s}(\mathbf{x})) \tag{1}$$

where $h_{\theta_p}(\mathbf{x})$ is a high-level feature tensor, $h_{\theta_s}(\mathbf{x})$ produces a corresponding mask, $\odot$ denotes element-wise multiplication, and a final classification head $g_\phi(\cdot)$ produces the logits $\mathbf{y}$.

**Our approach (Early Masking)** ensures faithfulness by applying the selector *before* the predictor. The selector model, $h_{\theta_s}(\cdot)$, produces an input-level binary mask $\mathbf{s} \in \{0, 1\}^{H \times W}$. The predictor network, $h_{\theta_p}(\cdot)$, is then architecturally constrained to only process information from the unmasked regions of the input image $\mathbf{x}$. This guarantees that the predictor's receptive field is strictly determined by the selector's mask.

**Implementation on a ViT with attention masks.** For a ViT-based predictor $h_{\theta_p}$, this input-level masking is implemented by modulating the self-attention mechanism in each layer. Rather than a simple element-wise multiplication on the input, we control which tokens can exchange information.

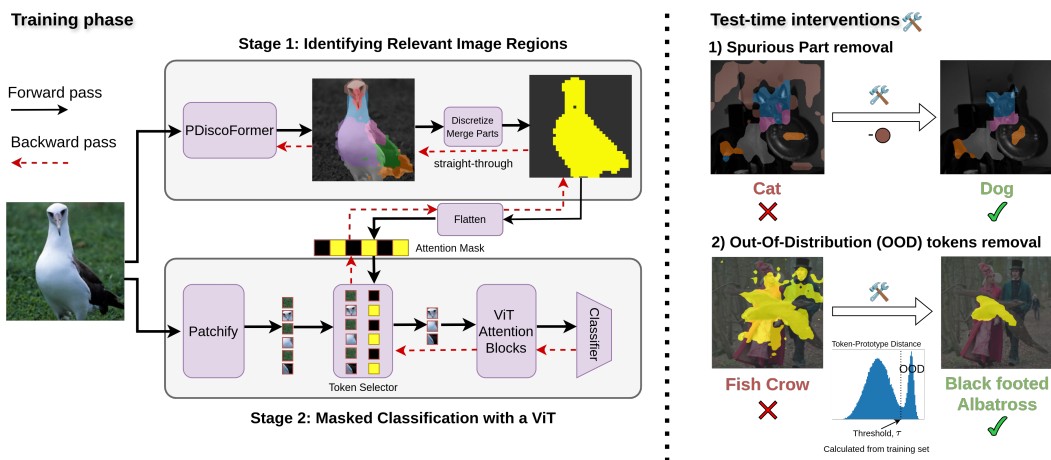

Figure 2: **Left:** iFAM first discovers task-relevant regions (Stage 1) and then classifies using only the selected regions (Stage 2), preventing reliance on background cues. **Right:** At test time, we leverage the model's inherently faithful region attribution to design (training-free) intervention strategies that further enhance robustness to spurious correlations.

Given the binary token selection mask $\mathbf{s} \in \{0, 1\}^N$ derived from the input mask, we construct an attention mask $\mathbf{M} \in \mathbb{R}^{N \times N}$:

$$\text{Attention}(\mathbf{Q}, \mathbf{K}, \mathbf{V}) = \text{softmax}\left(\frac{\mathbf{Q}\mathbf{K}^\top}{\sqrt{D}} + \mathbf{M}\right)\mathbf{V}, \tag{2}$$

where the elements in $\mathbf{M}$ are defined as:

$$M_{ij} = \begin{cases} -\infty, & \text{if } s_i = 0 \text{ or } s_j = 0 \\ 0, & \text{otherwise.} \end{cases} \tag{3}$$

This forces attention to and from masked-out tokens to be zero, preventing any information leakage. The final prediction $\mathbf{y}$ is then computed by applying the classification head to the output of the masked predictor: $\mathbf{y} = g_\phi(h_{\theta_p}(\mathbf{x}, \mathbf{s}))$.

## 3.2 STAGE 1: IDENTIFYING RELEVANT IMAGE REGIONS

To identify relevant image regions for the downstream task, we leverage the PDiscoFormer part discovery method Aniraj et al. (2024). This approach, guided solely by image-level class labels and part-shaping priors, partitions the image into $K + 1$ regions: $K$ distinct foreground parts plus the background, which is discarded. The discovered parts are shared across classes. Each part is associated with a learned prototype, encouraging semantic consistency across the dataset. The prototypes are also trained to be mutually de-correlated, so that each part captures a distinct aspect of the object. To this end, we use the original PDiscoFormer default settings.

## 3.3 STAGE 2: MASKED-INPUT CLASSIFICATION

PDiscoFormer suffers from the same issues that we have identified as flaws in attention mechanisms: it uses soft attention masks that are applied to a high-level representation. To address this drawback, we propose to make the masks binary, via discretization, and to use them to explicitly define the receptive field of the second stage model, using Eq. (2).

**Discrete masks.** PDiscoFormer produces part attention maps that assign, for each image token, a weight distribution across parts, with weights summing to one. These weights are designed to approach a hard assignment via Gumbel softmax, where one part receives a weight close to one, while the others are close to zero. However, we emphasize that these maps still remain a soft distribution across parts. This may seem as a subtlety, but we argue that only a truly discrete attribution map can provide faithfulness guarantees by fully preventing information leakage. To tackle this issue, we

introduce a discretization step for the obtained part maps prior to the second stage. At this point, the foreground parts are merged together to obtain a binary input mask for the second stage model. With the aim to allow gradient flow between the second and first stages, we employ the straight-through gradient trick used by Gumbel softmax Jang et al. (2017), where the hard masks are used in the forward pass and the soft ones in the backward pass.

**Input image masks.** An additional requirement in order to prevent information leakage, related to the receptive fields of modern computer vision architectures, is to adopt early masking Aniraj et al. (2023). That is, masking directly the input of the model instead of doing so at a higher-level representation. In this way, only the unmasked tokens are considered by the ViT, thus eliminating any possible information contamination from the unattended regions. To mitigate potential impacts on training dynamics from removing background tokens, our ViT architecture also incorporates register tokens, which are restricted to attend only to the foreground.

### 3.4 AUDITABLE ROBUSTNESS VIA TEST-TIME INTERVENTIONS

A key benefit of our framework is its inherent transparency, setting it apart from end-to-end black-box models. The explicit, semantically consistent part masks are auditable, enabling a novel form of human-AI collaboration: targeted, user-driven interventions at test time. We demonstrate this unique capability with two complementary methods:

**Drop a part that captures a spurious object.** While the Stage-1 selector is designed to discover the most informative, causal image regions, setting the number of parts $K$ too high can cause some parts to learn spurious correlations. A key feature of our auditable framework is the ability to mitigate this at test time. iFAM allows a user to *select a subset of the discovered parts* to feed into the Stage-2 classifier. Since the learned parts are shared across classes and semantically consistent across the dataset, this intervention can be performed globally. For instance, a user can manually inspect a few images to identify a part that consistently captures a spurious element and exclude it from the second stage (see Appendix D). Importantly, *this process can be automated*: a leave-one-out (LOO) analysis on a validation set can programmatically identify and exclude parts whose removal consistently improves per-class performance, offering a scalable solution.

**Drop tokens assigned to a part with low confidence.** In cases where OOD objects present at inference time lead to false positive part detections, it is possible to simply remove the low confidence tokens from any given part. This can be achieved by checking whether the assigned parts are unexpectedly distant from the corresponding prototype in the feature space, based on statistics drawn from the training set Liu et al. (2020). Specifically, a distance-based threshold $\tau_k^q$ can be calibrated on the training set given a large percentile $q$, such that $q$ is the proportion of tokens assigned to part $k$ that have a distance to the corresponding part prototype smaller than $\tau_k^q$. At inference, tokens assigned to part $k$ with distance exceeding $\tau_k^q$ are reclassified as background.

Finally, since these two approaches are complementary, the first addressing part-level intervention while the second covers individual tokens from all parts, they can be adopted simultaneously.

## 4 EXPERIMENTAL SETUP

We evaluate our method's ability to discover task-relevant regions and ignore spurious correlations using only image-level class labels. To do so, we test on a diverse suite of benchmarks specifically designed with known biases, spanning binary, fine-grained, and large-scale classification challenges. Implementation details are provided in Appendix A.

**Datasets and Evaluation Metrics.** To thoroughly assess robustness, our evaluation begins with two binary classification tasks with strong background correlations. In **MetaShift cat vs. dog** Liang et al. (2022); Wu et al. (2023), dogs (resp. cats) predominantly appear in outdoor (resp. indoor) environments during training, while the test set contains only indoor backgrounds, making dogs harder to detect. Similarly, in **Waterbirds** Sagawa et al. (2020), derived from CUB Wah et al. (2011), training set presents a 95% correlation between species (*waterbird/landbird*) and background (water/land), with the hardest test groups consisting of waterbirds on land and landbirds on water. We extend this evaluation to more complex scenarios, including the 200-way fine-grained task **CUB** evaluated on **Waterbird200**'s adversarial backgrounds and the medical dataset **SIIM-ACR** Zawacki et al. (2019), where artifacts such as chest tubes can bias pneumothorax detection Saab et al. (2022). Finally, to

Table 1: Results on MetaShift, Waterbird, ImageNet-1K (IN-1K), and IN-9 (Original: IN-9O; Mixed-Same: MS; Mixed-Rand: MR). BG-GAP = MS − MR (lower is better). Shaded columns: robustness metrics. [†] models trained with extra supervision; [‡] larger-capacity models. $K$: number of foreground parts. LLE: Last Layer Ensemble Li et al. (2023), SWAG Singh et al. (2022), MAE He et al. (2022), ❄: Frozen backbone, ♨ : Fine-tuned backbone, ✄ : Intervention, gt: Ground Truth Masks, f: FOUND (Saliency detection) Siméoni et al. (2023), [1]: SWAG Singh et al. (2022) pre-train + LLE Li et al. (2023), [2]: MAE He et al. (2022) pre-train + LLE Li et al. (2023), R-50: ResNet50, R-152: ResNet152.

| Method | Arch. | MetaShift | | | Waterbird | | |
|---|---|---|---|---|---|---|---|
| | | K | AA | WGA | K | AA | WGA |
| Early mask[gt†] (upper bound) | ViT-B | – | – | – | 1 | 99.2 | 97.2 |
| Late mask[gt†] (upper bound) | ViT-B | – | – | – | 1 | 95.7 | 84.0 |
| ERM Wu et al. (2023) | R-50 | – | 72.9 | 62.1 | – | 97.0 | 63.7 |
| ERM | ViT-B | – | 75.8 | 62.5 | – | 95.0 | 80.7 |
| DinoV2 ❄ | ViT-B | – | 83.2 | 72.6 | – | 95.9 | 88.5 |
| DinoV2 PCA Darbinyan et al. (2023) | ViT-B | – | – | – | – | 97.4 | 94.0 |
| DinoV2 ♨ | ViT-B | – | 84.7 | 76.8 | – | 98.6 | 95.8 |
| MaskTune Asgari et al. (2022) | R-50 | – | – | – | – | 93.0 | 86.4 |
| GroupDRO Sagawa et al. (2020) | R-50 | – | 73.6 | 66.0 | – | 91.8 | 90.6 |
| DISC Wu et al. (2023) | R-50 | – | 75.5 | 73.5 | – | 93.8 | 88.7 |
| PDiscoFormer Aniraj et al. (2024) | ViT-B | 2 | 86.9 | 81.0 | 4 | 96.0 | 87.4 |
| PDiscoFormer Aniraj et al. (2024) | ViT-B | 4 | 83.2 | 75.5 | 8 | 94.2 | 84.3 |
| Late mask[f] Siméoni et al. (2023) | ViT-B | 1 | 82.3 | 73.5 | 1 | 95.3 | 83.3 |
| Early mask[f] Siméoni et al. (2023) | ViT-B | 1 | 84.5 | 77.1 | 1 | 98.6 | 95.2 |
| iFAM | ViT-B | 2 | **89.1** | 86.3 | 4 | 98.7 | 96.4 |
| iFAM | ViT-B | 4 | 88.7 | **88.6** | 8 | **99.0** | **97.0** |

| (b) Results on ImageNet-9 (IN-9) Backgrounds Challenge | | | | | | |
|---|---|---|---|---|---|---|
| Method | Arch. | IN-1K | IN-9O | MS | MR | BG-GAP ↓ |
| ERM Wightman et al. (2021) | R-50 | 81.2 | 96.4 | 90.0 | 84.6 | 5.4 |
| ERM [‡] Wightman et al. (2021) | R-152 | 83.5 | 97.3 | 92.1 | 87.4 | 4.7 |
| ERM Touvron et al. (2022) | ViT-B | 83.8 | 97.9 | 92.4 | 87.9 | 4.6 |
| ERM [‡] Touvron et al. (2022) | ViT-L | 84.8 | 98.0 | 93.0 | 89.4 | 3.6 |
| DinoV2 Darcet et al. (2024) | ViT-B | 84.6 | 98.1 | 93.1 | 87.1 | 6.0 |
| DinoV2 [‡] Darcet et al. (2024) | ViT-L | **86.7** | 98.3 | **95.5** | 90.2 | 5.3 |
| MaskTune Asgari et al. (2022) | R-50 | - | 95.6 | 91.1 | 78.6 | 12.5 |
| LLE Li et al. (2023) | R-50 | 76.3 | 95.5 | 88.3 | 83.4 | 4.9 |
| SWAG+LLE[1] Li et al. (2023) | ViT-B | 85.2 | 98.0 | 92.4 | 87.9 | 4.5 |
| MAE+LLE[2] Li et al. (2023) | ViT-B | 83.7 | 97.4 | 92.5 | 88.3 | 4.2 |
| MAE+LLE [‡2] Li et al. (2023) | ViT-L | 85.8 | 97.4 | 93.5 | 89.8 | 3.6 |
| PDiscoFormer (K=1) Aniraj et al. (2024) | ViT-B | 83.3 | **98.4** | 93.9 | 88.6 | 5.3 |
| iFAM (K=1) | ViT-B | 84.3 | 97.5 | 93.5 | **91.1** | **2.4** |

demonstrate scalability, we use the **ImageNet-9 (IN-9) Backgrounds Challenge** Xiao et al. (2021). This benchmark measures background reliance via the **BG-GAP**, which is the accuracy drop when evaluating on images with same-class (**Mixed-Same**) versus random-class (**Mixed-Rand**) backgrounds. Across all relevant datasets, we report standard **average accuracy (AA)** alongside crucial robustness metrics, such as **worst group accuracy (WGA)** and the BG-GAP.

**Baselines.** We compare our method against several baselines, including the late-masking PDisco-Former Aniraj et al. (2024), standard CNN/ViT models, and specialized de-biasing methods. We also evaluate against saliency-based masking techniques Siméoni et al. (2023) and report results from models trained with extra supervision (e.g., ground-truth masks) as **upper bounds**.

## 5 RESULTS AND DISCUSSION

### 5.1 RESULTS ON ROBUSTNESS BENCHMARKS

Results in Tables 1 and 2 show that our two-step approach, which explicitly limits the predictor's receptive field to the discovered foreground regions, leads to significant improvements in robustness on datasets with spurious background correlations. Qualitative results are provided in Appendix D.

Table 2: Results on CUB, Waterbird200 (CUB with OOD backgrounds) and SIIM-ACR. Shaded columns: robustness metrics. [†] models trained with extra supervision . ❄ : Frozen backbone, ♠ : Fine-tuned backbone, ✗ : Intervention, AUC: Area Under the Curve, seg : Supervised Semantic Segmentation

**(a) Results on CUB and Waterbird200**

| Method | K | CUB in-distrib. | Waterbird200 OOD |
|---|---|---|---|
| Early mask[seg] [†] Aniraj et al. (2023) (upper bound) | 1 | 91.4 | 88.8 |
| Late mask[seg] [†] Aniraj et al. (2023) (upper bound) | 1 | 90.7 | 74.8 |
| ViT-B DinoV2 ❄ | - | 89.2 | 76.6 |
| ViT-B DinoV2 ♠ | - | **91.6** | 68.4 |
| PDiscoFormer Aniraj et al. (2024) | 4 | 89.1 | 76.0 |
| PDiscoFormer Aniraj et al. (2024) | 8 | 88.8 | 76.8 |
| PDiscoFormer Aniraj et al. (2024) | 16 | 88.7 | 75.8 |
| iFAM | 4 | 90.1 | 86.1 |
| iFAM | 8 | 90.4 | **86.2** |
| iFAM | 16 | 90.6 | **86.2** |

**(b) Results on SIIM-ACR**

| Method | K | A. AUC | WG AUC |
|---|---|---|---|
| BBox-ERM [†] Saab et al. (2022) (upper bound) | - | 92.4 | 72.0 |
| Seg-ERM [†] Saab et al. (2022) (upper bound) | - | 93.3 | 82.0 |
| ResNet50 Saab et al. (2022) | - | 90.9 | 45.5 |
| ResNet50 JTT Liu et al. (2021) | - | **92.6** | 55.9 |
| ResNet50 GEORGE Sohoni et al. (2020) | - | 92.0 | 63.4 |
| ViT-B RAD-DINO ❄ | - | 90.6 | 40.6 |
| ViT-B RAD-DINO ♠ | - | **92.6** | 54.3 |
| PDiscoFormer Aniraj et al. (2024) | 8 | **92.6** | 48.1 |
| iFAM | 8 | 92.1 | **65.9** |

**Results on MetaShift and Waterbird.** Tab. 1-a highlights the advantage of using a pretrained DINOv2 backbone, as also noted by Darbinyan et al. (2023). Notably, simply fine-tuning DINOv2 surpasses all prior OOD robustness methods, while the same ViT-B pretrained on ImageNet does not, underscoring the impact of self-supervised pretraining. Additionally, early masking consistently outperforms late masking in robust accuracy, whether using ground-truth masks or saliency-based selection Siméoni et al. (2023). Our method significantly improves upon these baselines, improving WGA from 81.0% to 88.6% on MetaShift and from 94.0% to 97.0% on Waterbird, effectively halving the error. Only early masking with ground-truth segmentation surpasses our results.

**Results on IN-9.** Tab. 1-b presents background sensitivity using the BG-GAP metric, which quantifies the accuracy difference between the Mixed-Same and Mixed-Rand variants. Surprisingly, vision transformers (ViTs) with advanced pre-training, such as DINOv2 Oquab et al. (2023); Darcet et al. (2024), perform worse than standard CNNs and ViTs trained purely on IN-1K following modern training protocols Touvron et al. (2022); Wightman et al. (2021), suggesting that pre-training does not inherently improve background robustness. While ResNets trained with de-biasing methods Li et al. (2023); Asgari et al. (2022) show slightly improved BG-GAP, they perform significantly worse on individual IN-9 variants, and ViTs with post-pretraining de-biasing objectives Li et al. (2023) offer only marginal gains. In contrast, our **iFAM** model achieves the lowest BG-GAP of **2.4**, outperforming its baseline (PDiscoFormer) and all other models, including larger architectures like ViT-L and ResNet152, which demonstrates the gains stem from our design rather than model capacity. We use only $K = 1$ for part-discovery methods due to the lack of common semantic parts across classes in this dataset.

**Results on CUB and Waterbird200.** Tab. 2-a shows that fine-tuning a DINOv2 ViT-B backbone does not scale well to fine-grained tasks. The fine-tuned CUB baseline underperforms its frozen counterpart on Waterbird200, despite a 2% in-distribution gain, suggesting overfitting to background cues. All late-masking models, including PDiscoFormer, saturate around 76% on Waterbird200, indicating that background biases persist even with an oracle late mask. Despite using only self-discovered masks, our method achieves 86.2%, closely matching early-masked models from Aniraj et al. (2023), which rely on supervised segmentation masks.

**Results on SIIM-ACR.** For SIIM-ACR (Tab. 2-b), training RAD-DINO or PDiscoFormer with late

Table 3: Results of applying the token removal intervention on MetaShift, Waterbird, SIIM-ACR, and the OOD Waterbird200 dataset.

| Method | MetaShift (K=8) | | Waterbird (K=16) | | SIIM-ACR (K=8) | | Waterbird200 (OOD) | | |
| | AA | WGA | AA | WGA | A. AUC | WG AUC | K=4 | K=8 | K=16 |
|---|---|---|---|---|---|---|---|---|---|
| iFAM | 84.5 | 78.8 | **98.8** | 97.0 | 92.1 | 65.9 | 86.1 | 86.2 | 86.2 |
| ✂ $q=97\%$ | **+0.2** | +0.3 | -0.1 | -0.4 | -0.1 | +0.1 | **+0.7** | +0.5 | **+1.1** |
| ✂ $q=99\%$ | **+0.2** | **+1.3** | 0.0 | **+0.4** | **+0.1** | **+0.5** | 0.5 | **+0.7** | 0.7 |

masking alone results in a biased model that overly relies on spurious correlations, leading to a WG AUC close to random performance. However, our method, with $K = 8$, achieves 65.9% WG AUC. This result can be further improved to 69.0% after interventions (see sec. 5.2), approaching the 72.0% obtained with ground-truth bounding boxes, despite not using such additional annotations.

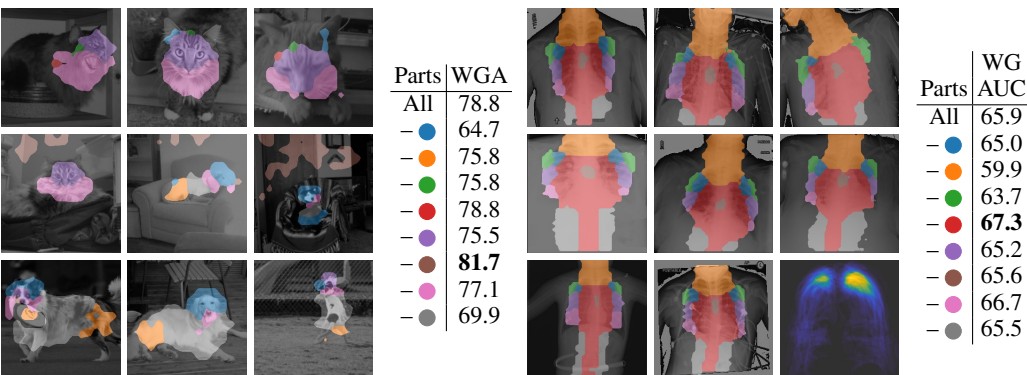

Figure 3: Leave-one-out (LOO) part removal intervention results on MetaShift (left) and SIIM-ACR (right) for $K = 8$. The bottom right image shows a heatmap of the average pneumothorax occurrence across the dataset.

## 5.2 ADDITIONAL ROBUSTNESS VIA INTERVENTIONS

In this experiment, we assess the impact of our intervention strategies on robustness to spurious correlations. Due to the weakly supervised nature of part discovery, our model may (i) identify spurious parts in datasets with stronger spurious correlations (e.g., MetaShift, SIIM-ACR), which can be tackled via **leave-one-out (LOO)** or (ii) assign out-of-distribution (OOD) objects to the foreground (e.g., Waterbird200), which we tackle via unconfident token removal.

**Part-Removal Intervention on MetaShift.** Fig. 3 (left) presents part assignment maps in MetaShift when using a too large number of parts, $K = 8$, color-coded, alongside WGA results from leave-one-out (LOO) evaluation. Most parts consistently capture coherent semantics. However, the brown part captures indoor elements, likely due to correlations between indoor backgrounds and the *cat* class. This demonstrates the power of the model's auditability: by inspecting the parts and removing this single spurious one, a user can steer the model to improve WGA from 78.8% to 81.7%, whereas removing other parts either reduces performance or has no effect.

**Part-Removal Intervention on SIIM-ACR.** Fig. 3 (right) shows SIIM-ACR results. Each part learns to focus on a different area of the torso. Removing the red part increases WG AUC by nearly 1.5 points. This part predominantly covers the central chest region, which has little overlap with common pneumothorax locations (see the heat map of average pneumothorax occurrence), but often contains spurious cues, such as drainage tubes.

**Token Removal Intervention.** Tab. 3 shows that this intervention consistently results in better OOD performance for all datasets, while the in-distribution performances are maintained. We also provide a more detailed analysis of this intervention in Appendix C where we study its effects on foreground discovery and the part assignment consistency in OOD settings.

**Combining interventions.** In Tab. 4 we also explore combining both intervention strategies and observe that they are highly complementary, leading to over four point improvement on MetaShift to 83.2% WGA when using $K = 8$, up from 78.8%, and over three points in SIIM-ACR, leading to

Table 4: Results on MetaShift and SIIM-ACR with combined interventions with $K = 8$.

| Method | MetaShift | | SIIM-ACR | |
| --- | --- | --- | --- | --- |
| | AA | WGA | A. AUC | WG AUC |
| PDiscoFormer Aniraj et al. (2024) | 83.2 | 75.5 | **92.6** | 48.1 |
| ✂ LOO | 85.2 | 76.8 | 92.6 | 48.1 |
| ✂ LOO + $q = 99\%$ | **85.4** | 76.8 | 92.6 | 48.2 |
| iFAM | 84.5 | 78.8 | 92.1 | 65.9 |
| ✂ LOO | 84.7 | 81.7 | 90.6 | 67.3 |
| ✂ LOO + $q = 99\%$ | 84.8 | **83.0** | 91.1 | **69.0** |

Table 5: Ablation results with $K = 4$.

| | CUB | Waterbird200 | MetaShift | |
| --- | --- | --- | --- | --- |
| | in-distrib. | OOD | AA | WGA |
| Full iFAM | 90.1 | **86.1** | **88.7** | **88.6** |
| No second stage | 89.1 | 76.0 | 83.2 | 75.5 |
| Soft masks | **90.6** | 85.7 | 88.0 | 86.3 |
| $K = 1$ w/o shaping (JAM) | 90.3 | 80.2 | 85.4 | 79.1 |
| No stage-1 classif. | 88.9 | 85.0 | 86.9 | 82.3 |
| Frozen stage-2 | 89.1 | 83.7 | 85.0 | 85.0 |

69% WG AUC, up from 65.9%. By contrast, PDiscoFormer benefits only marginally, with gains of about one WGA point on MetaShift and 0.1 WG AUC on SIIM-ACR.

## 5.3 ABLATION STUDIES

To understand the contribution of each component in our proposed method, we conduct an ablation study on the 200-way CUB/Waterbird200 benchmark and the binary MetaShift task (Tab. 5).

The results in Tab. 5 confirm that each component of our design is critical for OOD robustness. The most significant contribution comes from the two-stage architecture itself. Removing the second stage ("No second stage"), which reduces the model to a single-stage late-masking approach, causes the largest drop in OOD performance: WGA on MetaShift falls from 88.6% to 75.5%, and accuracy on Waterbird200 drops from 86.1% to 76.0%. Furthermore, replacing our discrete hard masks with continuous soft masks ("Soft masks") also degrades OOD performance, confirming that a strict removal of background tokens is necessary to prevent information leakage.

Crucially, we evaluate a variant where we remove the part-shaping losses ("$K = 1$ w/o shaping (JAM)"), reducing our selector to a standard Joint Amortized Model (JAM). The poor performance of this baseline (79.1% WGA on MetaShift and 80.2% on Waterbird200) provides direct empirical evidence that our semantic part-shaping objective is the key component that advances beyond prior selector-predictor architectures. Finally, ablating the Stage-1 classification loss and keeping Stage-2 frozen also reduce performance, validating our end-to-end joint training strategy.

## 6 CONCLUSION

**Limitations.** The main limitation of our approach is the extra computational cost incurred by the use of two forward passes: one for part discovery and the second for the downstream task. While the straight-through gradient requires the entire image to be processed during training, the second pass only requires access to a subset of the image at inference, allowing optimization via patch token pruning Li et al. (2022).

**Conclusion.** We investigated a two-step framework where stage 1 processes the full image to discover task-relevant regions, while stage 2 operates exclusively on this binary selection. By guaranteeing the receptive field of the stage 2 predictor through attention masking, we ensure that only the regions identified by stage 1 influence its representations, thereby minimizing background-related biases. Empirically, we show that this approach significantly improves robustness on benchmarks designed to test resilience against such biases. Our findings highlight the importance of inherently faithful attention mechanisms for developing robust computer vision systems.

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

## A  IMPLEMENTATION DETAILS

All models are implemented in PyTorch, using a ViT-B backbone Darcet et al. (2024) initialized with DINOv2 weights Oquab et al. (2023) (or RAD-DINO for SIIM-ACR Pérez-García et al. (2025)).

### A.1  TRAINING SETTINGS

We trained all models for 90 epochs using the AdamW optimizer Loshchilov & Hutter (2019). During the part discovery stage, we followed the procedure outlined in the original paper Aniraj et al. (2024). Specifically, the class token, position embedding, and register token were kept unfrozen, while the remaining ViT layers were frozen. In this stage, we trained these unfrozen tokens along with the randomly initialized layers, including the projection, modulation, and final classification layers. In the second stage, we fine-tuned all parameters of the model.

To adjust the learning rate dynamically, we employed a cosine annealing schedule Loshchilov & Hutter (2022). The initial learning rates were set as follows: $10^{-6}$ for the fine-tuned tokens of the ViT backbone in both stages and for the layers of the second-stage ViT, $10^{-3}$ for the linear projection layer forming the part prototypes, and $10^{-2}$ for the modulation and final linear layers used for classification in both stages.

We used a variable batch size, with a minimum of 16, depending on the available computational resources. To scale the learning rate appropriately, we applied the square root scaling rule Krizhevsky (2014). Regularization was performed using gradient norm clipping Pascanu et al. (2013) with a constant value of 2 and a normalized weight decay Loshchilov & Hutter (2019) set to 0.05.

The PDiscoFormer losses were configured as in the original paper Aniraj et al. (2024), with one exception for the biomedical dataset SIIM-ACR Zawacki et al. (2019). For this dataset, we disabled the background loss $\mathcal{L}_{p_0}$ by setting its weight to 0, as this loss assumes the background part is more likely to occur at the image boundaries — an assumption that does not necessarily hold for pneumothorax occurrences.

Finally, we used a constant part dropout value of 0.3 for both stages of the model in all experiments. The dropout value for the first stage aligns with that used in the original PDiscoFormer paper Aniraj et al. (2024), while the value for the second stage was ablated in Table 5 of our main paper.

**Scaling up to larger datasets.** For larger datasets such as ImageNet1K Russakovsky et al. (2015), we adopted optimizations including Automatic Mixed Precision (AMP) Micikevicius et al. (2018) and temporal averaging using Exponential Moving Average (EMA) Kingma (2015); Morales-Brotons et al. (2024) to accelerate and stabilize training. By leveraging these optimizations, we were able to double the batch size, leading to a $3.5\times$ reduction in training time, all while maintaining performance. Additionally, we found that larger datasets benefited from longer training, prompting us to increase the total number of epochs to 120.

**Baseline Training Settings.** Wherever possible, we report results from cited papers or evaluate public weights; otherwise, we re-train baselines using the experimental setup from the original paper.

## B  TRAINING TIME AND INFERENCE SPEED

We use an input image size of 518 for the CUB Wah et al. (2011), Waterbirds Sagawa et al. (2020), SIIM-ACR Zawacki et al. (2019) aligning with the default resolution of DINOV2. This higher resolution is consistent with prior works van der Klis et al. (2023); Aniraj et al. (2024); Saab et al. (2022). For the MetaShifts Liang et al. (2022) and ImageNet1K datasets, we adopt a reduced input size of 224, resulting in lower computational requirements.

**Training Time.** On a machine with 8 NVIDIA A100 GPUs, the training times are as follows: approximately 3 hours for CUB and Waterbirds, 5 hours for SIIM-ACR, 11 minutes for MetaShifts, and 34 hours for ImageNet-1K (with AMP and EMA optimizations).

**Inference Speed.** On an RTX 3090, models trained on CUB (input size: 518) run at 43 images/second, while those trained on MetaShift (input size: 224) reach 151 images/second. These results are reported without any inference-time optimizations. We believe future work can further improve speed by leveraging the sparsity of second-stage inputs.

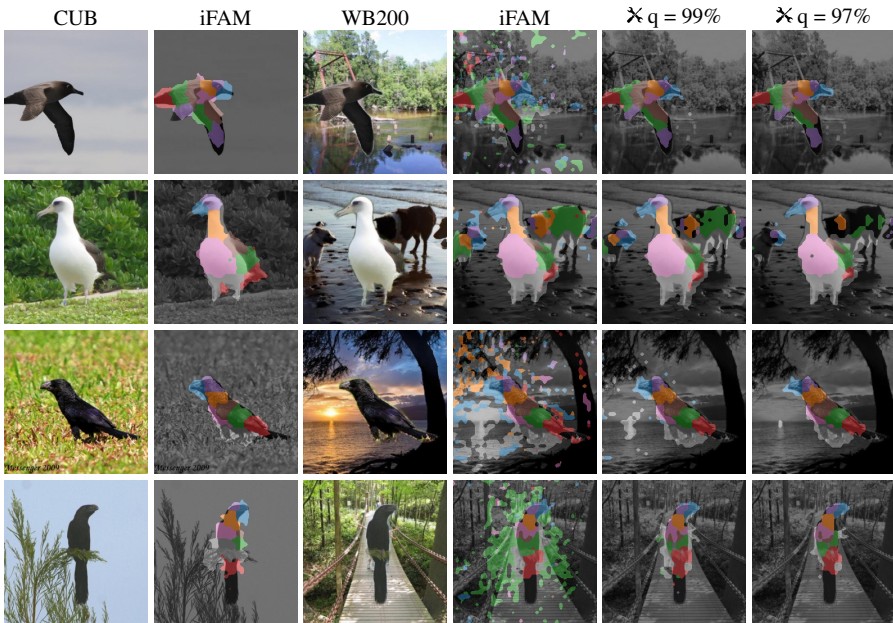

Figure 4: Qualitative results of part discovery of our model on the CUB dataset ($K = 8$), along with results on the corresponding out-of-distribution (OOD) images from the WB200 (WaterBirds200) dataset and the effect of the test-time intervention of thresholding on the OOD images.

## C  DETAILED ANALYSIS OF TOKEN REMOVAL

### C.1  QUALITATIVE ANALYSIS

Fig. 4 illustrates OOD token removal for $K = 8$. In CUB (second column), discovered parts align well with the bird. However, in Waterbird, background objects are often misassigned to foreground parts. Since these objects have representations farther away from part prototypes, applying a $97^{th}$ percentile threshold effectively removes them. This results in a small but consistent improvement in Waterbird200 (Tab. 3), with over a one-point gain at $K = 16$.

### C.2  QUANTITATIVE ANALYSIS

In Table 3, we demonstrated that the OOD token removal intervention consistently improves classification accuracy. To provide a deeper analysis, we now evaluate its effect on part assignment consistency and foreground discovery capability using the metrics defined below.

**Evaluation Metrics.** The CUB dataset provides ground-truth annotations for parts in the form of keypoints, which denote the centroid locations of parts within each image, as well as foreground-background masks. Since the images in the Waterbird200 dataset are identical to those in CUB, differing only in their adversarial backgrounds, the CUB annotations can also be used for Waterbird200. We evaluate foreground discovery using **mean Foreground Intersection-over-Union (Fg. mIoU)** and part assignment consistency using **Keypoint Regression (Kp)**.

1. **Fg mIoU.** This metric assesses the model's ability to identify the foreground region relevant for downstream classification. We merge all detected foreground parts and compute the IoU between the merged parts and the ground-truth foreground-background masks from the CUB dataset.

2. **Kp.** Following Hung et al. (2019), we measure part assignment consistency by deriving landmark locations through a trained linear regression model. This model maps the 2D geometric centers of the part assignment maps to their corresponding ground-truth part landmarks. The predicted landmarks are then compared against ground-truth annotations on the test set, with the evaluation metric being the normalized mean L2 distance.

Table 6: Quantitative analysis of the effect of the token removal intervention on part assignment consistency using keypoint regression (Kp) and foreground discovery (Fg. MIoU) on the OOD Waterbird200 dataset. $K$: Number of foreground parts.

| Method | K | Kp $\downarrow$ | Fg. MIoU $\uparrow$ | Top-1 Acc. $\uparrow$ |
|---|---|---|---|---|
| iFAM | | 10.3 | 63.7 | 86.1 |
| ✂ $q = 97\%$ | 4 | **8.4** | 65.2 | **86.8** |
| ✂ $q = 99\%$ | | 9.2 | **65.9** | 86.6 |
| iFAM | | 9.3 | 68.6 | 86.2 |
| ✂ $q = 97\%$ | 8 | **6.7** | 71.4 | 86.7 |
| ✂ $q = 99\%$ | | 7.3 | **72.4** | **86.9** |
| iFAM | | 8.0 | 70.2 | 86.2 |
| ✂ $q = 97\%$ | 16 | **6.2** | 72.9 | **87.3** |
| ✂ $q = 99\%$ | | 6.5 | **73.1** | 86.9 |

**Results on Foreground Discovery.** The low-confidence token removal technique consistently improves Foreground MIoU across all values of $K$ on the OOD Waterbird200 dataset (see Tab. 6). However, increasing the threshold (e.g., ✂ $q = 97\%$) leads to a slight reduction in MIoU compared to using ✂ $q = 99\%$. For instance, at $K = 8$ (results shown in Figure 4 of the main paper), the baseline model achieves a Foreground MIoU of 68.6%, which improves to 72.4% with ✂ $q = 99\%$, but drops to 71.4% with ✂ $q = 97\%$, suggesting that a stricter confidence threshold may inadvertently remove some foreground regions. Despite this, the drop in classification accuracy is minimal (from 86.9% to 86.7%), indicating that the model remains robust to removed foreground regions. Similar trends are observed across other values of $K$, where ✂ $q = 99\%$ generally leads to the best Foreground MIoU, while ✂ $q = 97\%$ provides slightly better classification performance.

**Results on Part Assignment Consistency.** The intervention improves keypoint regression (Kp) values across all $K$ values, indicating that the centroids of part assignment maps align more closely with ground-truth annotations. For instance, at $K = 16$, the Kp value improves from 8% (baseline) to 6.2% (✂ $q = 97\%$), likely due to the removal of low-confidence tokens near part boundaries, as shown in Fig. 4.

Overall, these results suggest that low-confidence token removal enhances both foreground discovery and part assignment consistency, with ✂ $q = 99\%$ generally yielding the best Foreground MIoU, while ✂ $q = 97\%$ slightly improves classification performance.

# D    QUALITATIVE RESULTS FOR PART DISCOVERY

To complement the quantitative evaluations in the main paper, we provide additional qualitative results in Figures 5 to 10. These results demonstrate our model's ability to discover meaningful parts and accurately identify foreground regions, which are crucial for downstream classification tasks and improving model interpretability.

**Results on CUB and WaterBird.** In datasets such as CUB and Waterbird, where all images belong to a single super-class (birds), the granularity of the discovered parts improves as $K$ increases. The identified parts generally align well with the foreground regions, as shown in Fig. 5 and Fig. 6.

**Results on MetaShifts.** For the binary classification task in MetaShifts (Cat vs. Dog), illustrated in Fig. 7, the model assigns a single part (blue) to both cats and dogs when $K = 1$. At $K = 2$, the same part (orange) is assigned to both classes, while another part (blue) is allocated to objects that frequently co-occur with these animals in the training set. However, at higher values of $K$, such as $K = 8$, the model begins to identify more non-causal or spurious parts.

**Results on ImageNet-1K.** Qualitative results on ImageNet-1K for various animal classes, including birds, cats, dogs, and insects, are shown in Figures 8, 9, and 10 for $K = 1$. At this setting, the model effectively performs foreground discovery, which appears to generalize well across the 1000 classes of ImageNet. This observation aligns with our quantitative results on background robustness in Table 1-b of the main paper.

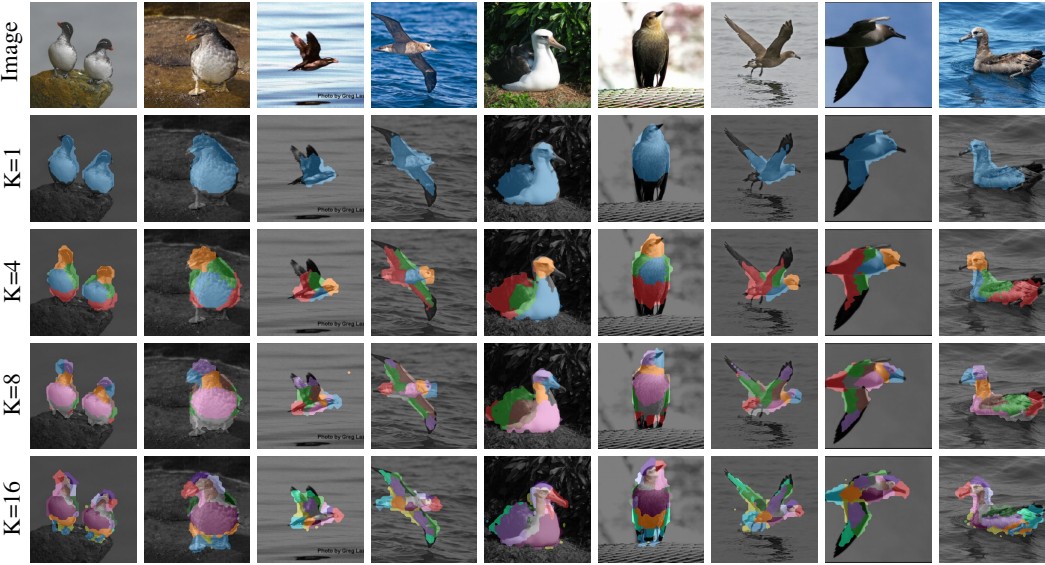

Figure 5: Qualitative results for part discovery for the iFAM model (without any ✂) trained on the CUB dataset for different values of K, the number of foreground parts.

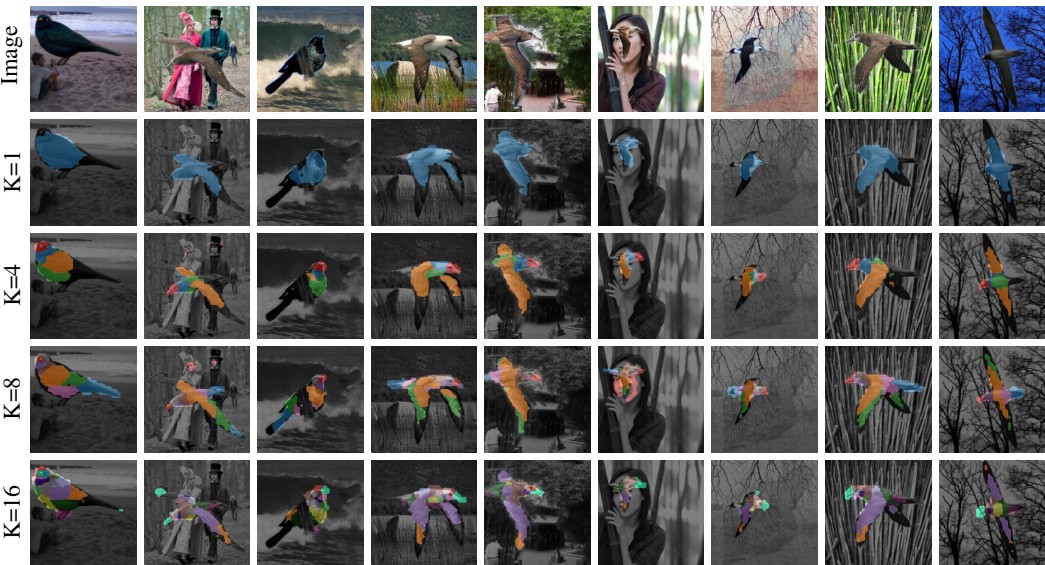

Figure 6: Qualitative results for part discovery for the iFAM model (without any ✂) trained on the Waterbirds dataset for different values of K, the number of foreground parts.

## E  USE OF LLMs IN WRITING

LLMs were used strictly for grammatical corrections and eventual rephrasing of the authors' original content.

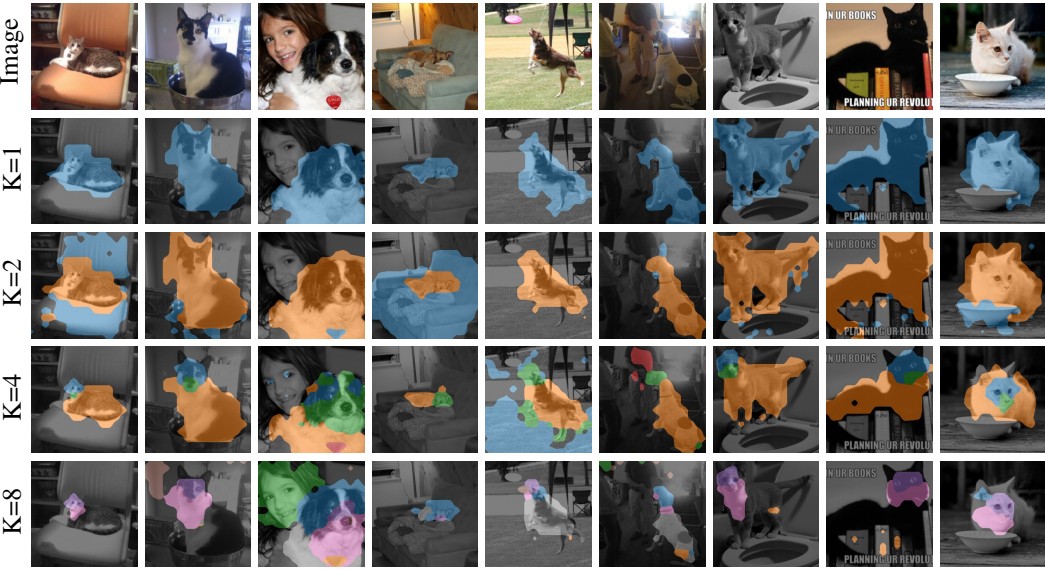

Figure 7: Qualitative results for part discovery for the iFAM model (without any ✂) trained on the MetaShifts dataset for different values of K, the number of foreground parts.

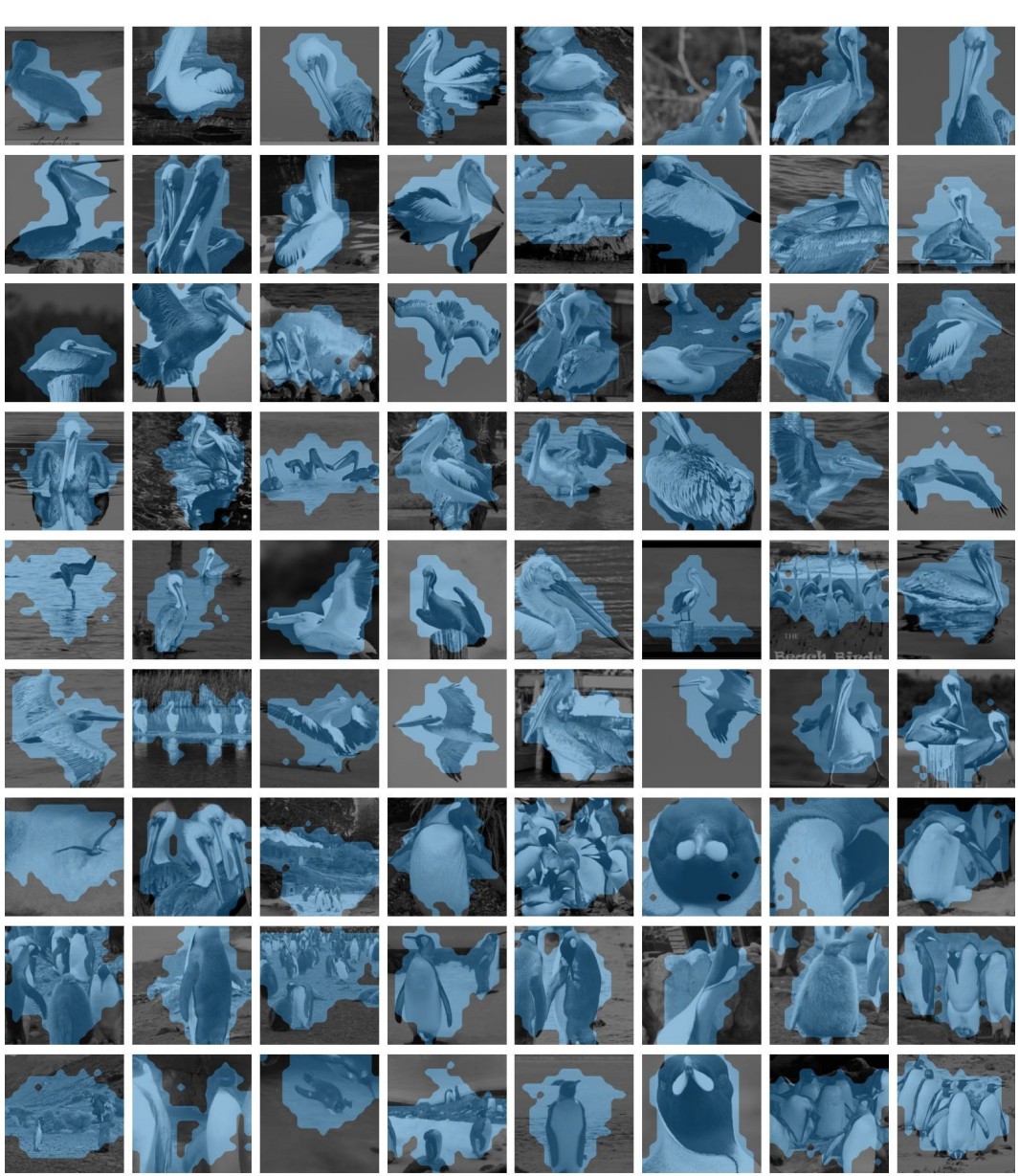

Figure 8: Qualitative Results on ImageNet-1K for Birds (without any ✂) for $K = 1$.

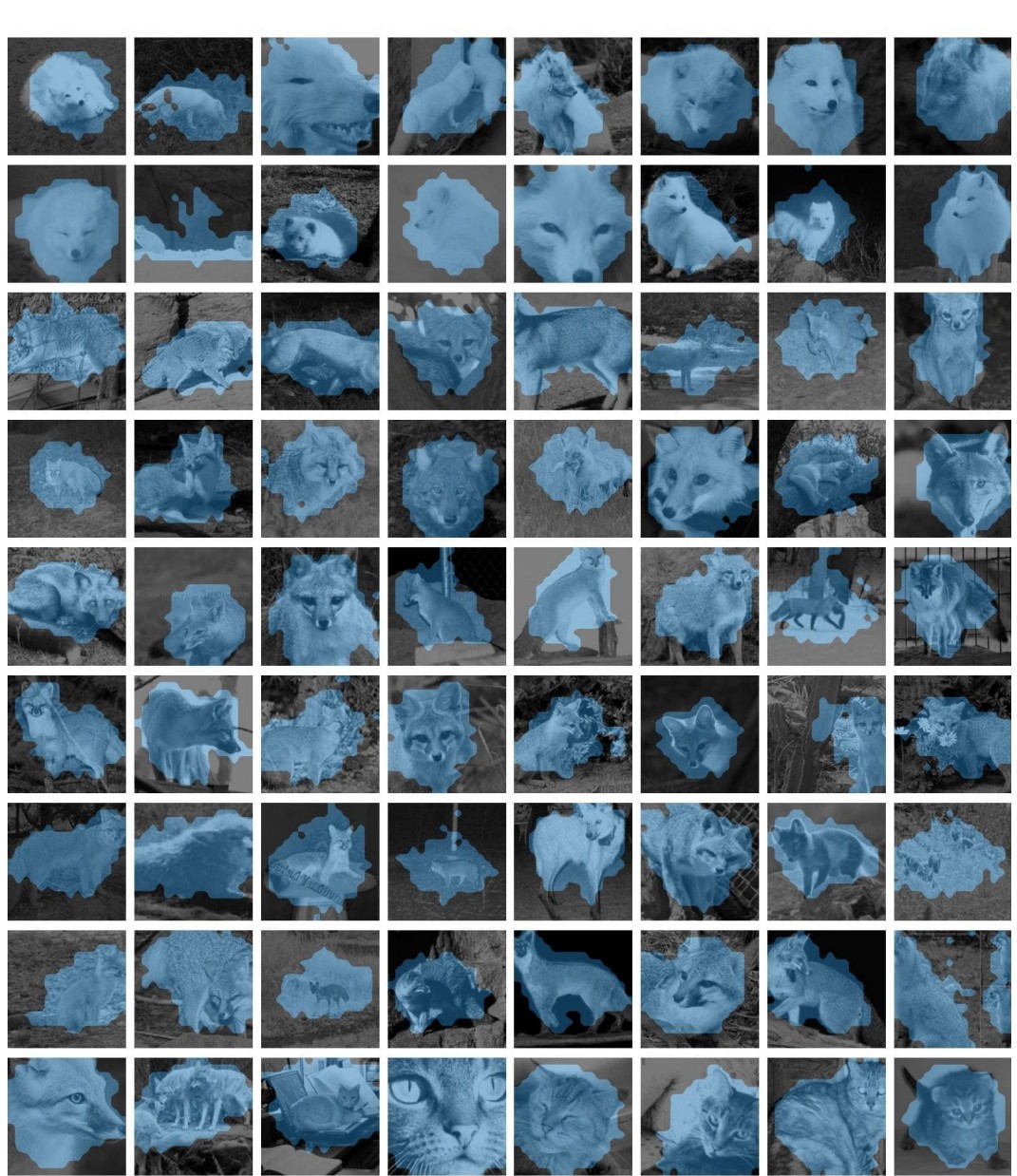

Figure 9: Qualitative Results on ImageNet-1K for Cats and Dogs (without any ✂) for $K = 1$.

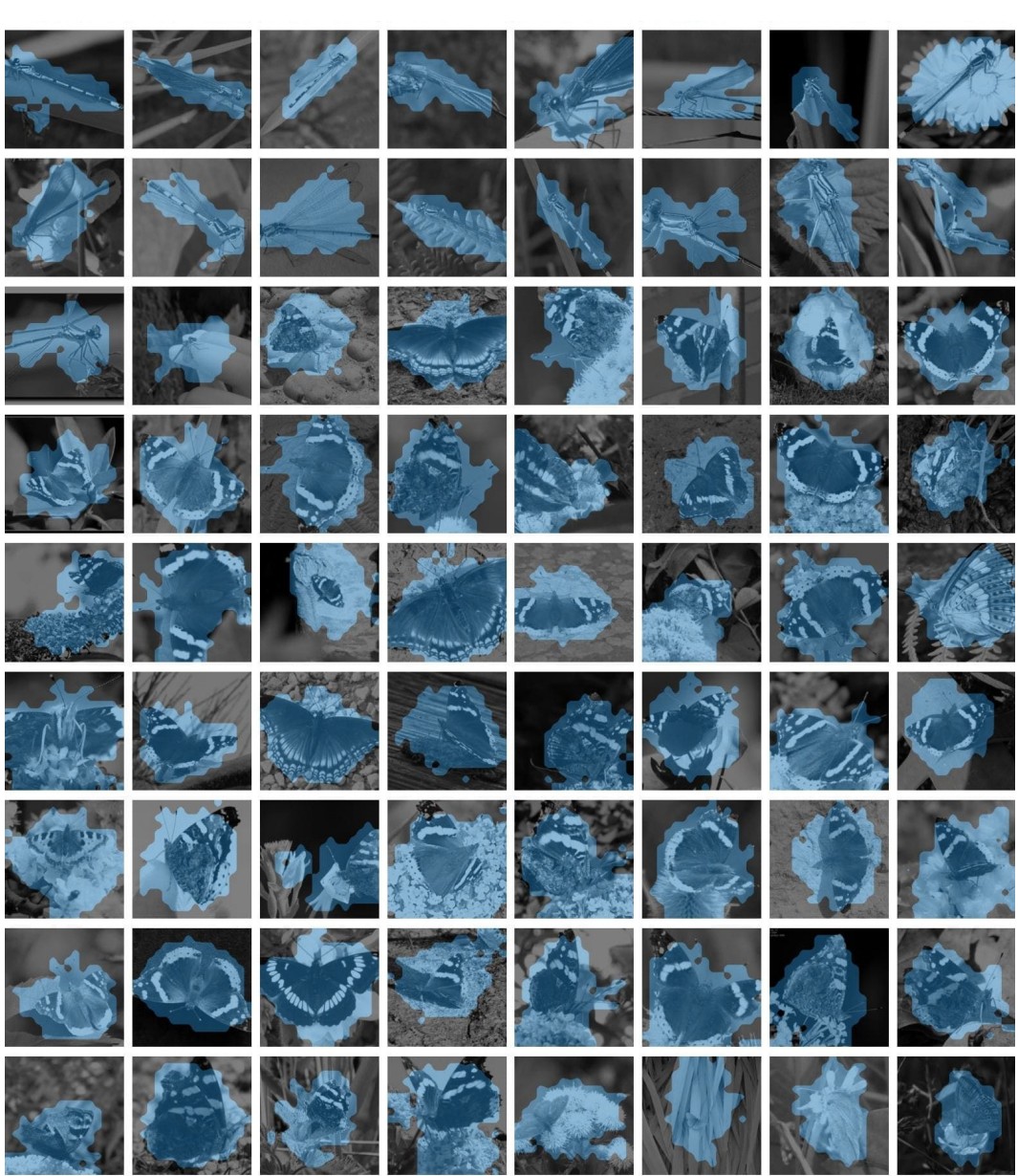

Figure 10: Qualitative Results on ImageNet-1K for Insects (without any ✂) for $K = 1$.

