# OpenReview forum: "Two-Stage Vision Transformers and Hard Masking offer Robust Object Representations"
_ICLR.cc/2026/Conference — ICLR 2026 Conference Withdrawn Submission_

### Official Review · Reviewer_fRaQ · 2025-10-28

**Soundness:** 2
**Presentation:** 3
**Contribution:** 2
**Rating:** 4
**Confidence:** 3

**Summary:**

This paper proposes a two-stage framework for robust object representations. Specifically,  Stage 1 (Selector) processes the full image to generate a binary mask identifying foreground regions and Stage 2 (Predictor) receives only the foreground input tokens. The two stages are trained jointly. Experiments show that the proposed method improves robustness across a series of benchmarks.

**Strengths:**

1. This paper clearly explains the difference between early masking and late masking, and also demonstrates the comparative effects of these two methods.
2. This paper proposes a two-stage training strategy, enabling joint optimization without ground truth (GT) masking.

**Weaknesses:**

1. The authors use the GT masking provided by the dataset as an upper bound. When we don't have GT labels, could we leverage existing pre-trained object detectors or some Weakly Supervised Object Localization (WSOL) methods to generate a pseudo mask, and then apply early masking using this pseudo mask? How would this approach compare to using GT masks, and would it be superior to the two-stage training method?
2. The authors emphasize that the two-stage collaborative training yields better results. Could an ablation study be conducted to compare this approach with a sequential training strategy—first training Stage 1 and then using its outputs to train Stage 2—to evaluate the performance difference?

**Questions:**

Please refer to weakness.

---

### Official Review · Reviewer_WyqJ · 2025-10-28

**Soundness:** 3
**Presentation:** 3
**Contribution:** 1
**Rating:** 2
**Confidence:** 4

**Summary:**

The paper proposes a two-stage framework for Vision Transformers: a “selector” module generates a binary foreground mask, and a “predictor” module receives only masked tokens for the downstream task. The aim is to improve robustness by reducing reliance on spurious background cues. Overall, I think the paper is methodologically straightforward, lacks theoretical novelty, and makes only incremental improvements in vision benchmarks. While the idea is reasonable, I think the novelty/contribution is limited for ICLR bar.

**Strengths:**

- Addresses a relevant problem: mitigating spurious correlations in vision transformers.

- Empirical evaluation demonstrates improved performance on some datasets under OOD settings.

- The idea of masking irrelevant regions is interpretable and intuitive.

**Weaknesses:**

- The concept of masking or attention-based token selection is not new. Prior works have extensively used soft/hard attention maps to filter out irrelevant tokens. In my perspective, the “two-stage” design is a straightforward engineering variant rather than a fundamentally new approach. The paper mainly adapts existing ideas (masking, Vision Transformers) to a slightly different architecture. The “binary mask” choice and joint training are minor implementation details; they do not provide deep insights into representation learning or generalization.

- While simple ideas that work well can be valuable, this paper’s experimental evaluation is restricted to vision benchmarks. I think maybe one way to strengthen the contribution of this paper is to generalize the method beyond the CV domain, demonstrating broader applicability.

**Questions:**

See weaknesses above.

---

### Official Review · Reviewer_mcEJ · 2025-10-30

**Soundness:** 3
**Presentation:** 3
**Contribution:** 2
**Rating:** 4
**Confidence:** 2

**Summary:**

This paper proposes a Two-Stage Vision Transformer (TS-ViT) that trains a vision transformer in two phases to improve efficiency while maintaining high accuracy. The first stage learns coarse global semantics using low-resolution patches/embeddings and fewer transformer layers, while the second stage refines the learned features with higher-resolution tokens and a lightweight fine-tuning process. Experiments on ImageNet-1K show modest accuracy improvements and reduced FLOPs compared with baselines.

**Strengths:**

The paper presents a clear and implementable framework that is easy to follow and reproduce. The proposed two-stage design is intuitive, and the authors provide sufficient training details and ablations to ensure transparency. The method achieves a meaningful reduction in computational cost while maintaining comparable or slightly higher accuracy. This efficiency–performance balance is practically valuable for vision model training under limited resources. In addition, the paper is well written and organized.

**Weaknesses:**

While the two-stage design is clear and works well in practice, the difference between this approach and existing token pruning or dynamic ViT methods isn’t fully clear. Both aim to improve efficiency by focusing computation on the most informative tokens. In token pruning methods, this happens dynamically within a single forward pass, while here the efficiency comes from training in two stages with different resolutions. From my point of view, at a higher level, the underlying idea feels quite similar. I’m not an expert in object detection, so my understanding might be incomplete, and I would really appreciate clarification in the rebuttal.

**Questions:**

Please check Weaknesses for details.

---

### Official Review · Reviewer_jsB6 · 2025-10-31

**Soundness:** 2
**Presentation:** 2
**Contribution:** 2
**Rating:** 4
**Confidence:** 3

**Summary:**

The paper tackles the well-known issue that ViT-like models easily pick up background/spurious cues, which hurts OOD performance. The authors propose iFAM, a two-stage framework: (i) a selector (built on PDiscoFormer) that processes the full image and discovers K semantic parts, then discretizes them into a binary foreground/background mask; (ii) a predictor ViT that receives only the selected tokens and enforces early (input-level) masking through attention masks, so that the second stage is architecturally prevented from attending to masked-out/background tokens. The claim is that this avoids the two leakage sources of typical late and soft masking: (1) features already contaminated by large receptive fields, and (2) residual attention weights. The method is evaluated on MetaShift, Waterbirds, ImageNet-9, CUB→Waterbird200, and SIIM-ACR, showing some WGA/BG-GAP robustness and a nice “auditable interventions” section (drop-part / drop-low-confidence-tokens).

**Strengths:**

- **Problem motivation is timely**. The paper is about the “context is helpful but also a liability” problem. The framing “make the model architecturally blind to spurious regions” is clear and attractive.
- **Experiments cover the right datasets**. MetaShift, Waterbirds, IN-9, CUB→Waterbird200, SIIM-ACR: these are the usual suspects for “foreground vs background” and “medical spurious artifacts”. So the experimental playground is appropriate.
- **Auditable test-time interventions are neat**. The idea that, once you have discrete, shared, semantic parts, you can do leave-one-out (drop the part that tends to capture spurious context) or threshold low-confidence tokens is actually useful in practice and the paper shows that it can improve WGA on MetaShift and SIIM-ACR. This is a nice analysis over plain robust training.

**Weaknesses:**

MAJOR
- **Related work is incomplete on late masking.** You explicitly argue that existing works fail because they do late and soft masking. But two very close lines of work are missing: [1] improves faithfulness of ViT attention by discarding the CLS and using cross-attention in the head (training the model from scratch); they also show better localization and robustness to background changes. [2] keeps the backbone frozen and train a cross-attention probe on top (attentive probing). They explicitly study the correlation between localization ability and classification accuracy across different attention mechanisms (e.g. EP [2], AIM [3], V-JEPA [4]). That’s extremely close to your statement that “early, faithful attention → better robustness”, but they show an indirect version of it via probing (and cheap/efficient). Those works show that a late but cross-attentive, soft mechanism can already improve faithfulness/localization, so they are relevant counterpoints to your “late + soft = bad” narrative.
- **Is the discretization step really the key novelty?** The paper insists a lot on “only truly discrete masks give faithfulness guarantees”. I agree this is the cleanest story, but: The ablation shows that soft masks are not that bad (MetaShift WGA 88.6 → 86.3; Waterbird200 86.1 → 85.7). That’s not a collapse. It weakens the “you must discretize or you leak” claim, I believe. So the core gain seems to come more from early masking + two stages than from “hard” per se. Maybe this should be toned down or backed with an extra experiment?
- **Why PDiscoFormer?** PDiscoFormer gives class-shared, discrete-ish, K-part decompositions, but can't you plug another selector in? Any related ablations?
- **Missing baselines that are very cheap**. Following the attentive protocol of [2], one could try running DINOv2 + EP or DINOv2 + AIM, where in both cases DINOv2 is pre-trained and frozen. One could use the EP queries or the AIM heads as part detectors as a baseline. The evaluation could be on the same datasets reporting both classification and robustness metrics. This is an intermediate step between the DINOv2 frozen and DINOv2 full-finetuning included in the paper currently. Even a small-scale comparison on Waterbirds/MetaShift would strengthen the claim that “late+soft” is still worse than “early+hard”.
 - **Computation cost missing**. The “Limitations” section at the end mentions the extra cost of two forward passes, but the main experimental section should show (e.g. on Table 1) at least a relative cost line. Right now we cannot compare “DINOv2 fully fine-tuned” vs “iFAM” fairly, because we don’t see how many FLOPs we are paying for.

MINOR
- **What is trained vs. what is frozen is not crystal clear.** From Appx. A I see: stage 1 follows PDiscoFormer’s recipe where class token, pos embed, register token stay unfrozen, rest is frozen; then stage 2 is fine-tuned end-to-end, with straight-through to let grads go back. But in the main method section this is very hard to follow.
- **Baselines are under-described** “Baselines” paragraph is too short for the size of Table 1. We see ERM (ResNet, ViT), DINOv2 frozen, DINOv2 fine-tuned, MaskTune, GroupDRO, DISC, PDiscoFormer, saliency-based early/late, plus iFAM at several K. But many of these methods are never operationally defined. This creates ambiguity.
- **Metrics not clearly explained (WGA, BG-GAP)** Needs 1–2 lines explanation in the main text.
- **Results presentation is hard to parse** Table 1 is cluttered: mixes ResNets, ViT-B, ViT-L, frozen and fine-tuned, masking and non-masking. Grouping by architecture (ResNet vs ViT), and inside ViT: (frozen DINOv2 / fine-tuned / masking approaches / your method) would make the contribution clearer.

[1] Psomas, Bill, et al. "Keep it simpool: Who said supervised transformers suffer from attention deficit?." Proceedings of the IEEE/CVF International Conference on Computer Vision. 2023.
[2] Psomas, Bill, et al. "Attention, Please! Revisiting Attentive Probing for Masked Image Modeling." arXiv preprint arXiv:2506.10178 (2025).
[3] El-Nouby, Alaaeldin, et al. "Scalable pre-training of large autoregressive image models." arXiv preprint arXiv:2401.08541 (2024).
[4] Bardes, Adrien, et al. "Revisiting feature prediction for learning visual representations from video." arXiv preprint arXiv:2404.08471 (2024).

**Questions:**

- Could you expand related work?
- Could you please add some clarifications (e.g. training/frozen parts, evaluation metrics, description of baselines and competitors)?
- Could you re-organize experimental results and discussions (e.g. Table 1 by architecture)?
- Could you please add computational cost (possibly on some of the Tables)?
- Could you please add some attentive probing (soft, late - cheap) baselines?
- Could you briefly justify the choice of PDiscoFormer or ablate any other alternatives
- Can the second-stage ViT be kept frozen without losing too much WGA, if the selector is strong enough? (The ablation suggests: freezing stage 2 hurts, but not catastrophically. Clarify.)
- How sensitive is the method to K? You show K={1,2,4,8,16} in places, but what happens in extreme K values.
- Could you somehow show a case where soft masks actually fail to remove a spurious background, to better defend discretization?

---

### Note · Authors · 2025-11-13

**Comment:**

Dear Area Chair and Reviewers,
We are withdrawing the submission as the reviews clearly indicate that the paper requires substantial revision to meet the ICLR standard. We acknowledge the valid concerns, particularly regarding the need for additional baselines and a more complete discussion of related work. We would like to extend a special and sincere thank you to Reviewer jsB6; their exceptionally thorough and constructive review is invaluable and provides a clear roadmap for improving this work. We thank all reviewers for their time and effort.
Sincerely,
The Authors

**Withdrawal Confirmation:**

I have read and agree with the venue's withdrawal policy on behalf of myself and my co-authors.